# ChAdOx1 nCoV-19 Immunogenicity and Immunological Response Following COVID-19 Infection in Patients Receiving Maintenance Hemodialysis

**DOI:** 10.3390/vaccines10060959

**Published:** 2022-06-16

**Authors:** Wisit Prasithsirikul, Tanawin Nopsopon, Phanupong Phutrakool, Pawita Suwanwattana, Piyawat Kantagowit, Wannarat Pongpirul, Anan Jongkaewwattana, Krit Pongpirul

**Affiliations:** 1Bamrasnaradura Infectious Diseases Institute, Nonthaburi 11000, Thailand; drwisit_p@yahoo.com (W.P.); pawitasuwan@gmail.com (P.S.); awannarat@yahoo.com (W.P.); 2Department of Preventive and Social Medicine, Faculty of Medicine, Chulalongkorn University, Bangkok 10330, Thailand; phanupong.dell@gmail.com (P.P.); kantpiya@windowslive.com (P.K.); 3School of Global Health, Faculty of Medicine, Chulalongkorn University, Bangkok 10330, Thailand; 4Harvard T.H. Chan School of Public Health, Harvard University, Boston, MA 02215, USA; 5National Center of Genetic Engineering and Biotechnology (BIOTEC), National Science and Technology Development Agency (NSTDA), Pathum Thani 12120, Thailand; anan.jon@biotec.or.th; 6Department of International Health, Johns Hopkins Bloomberg School of Public Health, Baltimore, MD 21205, USA; 7Bumrungrad International Hospital, Bangkok 10110, Thailand

**Keywords:** COVID-19, SARS-CoV-2, ChAdOx1 nCoV-19, hemodialysis, ESRD, vaccines, immunogenicity, post-infection

## Abstract

Patients with end-stage renal disease (ESRD) receiving hemodialysis (HD) were found to have a decreased immune response following mRNA COVID-19 immunization. ChAdOx1 nCoV-19 was a promising COVID-19 vaccine that performed well in the general population, but the evidence on immunogenicity in ESRD with HD patients was limited. Moreover, the immunological response to COVID-19 infection was inconclusive in patients with ESRD and HD. The aim of this study was to investigate the immunogenicity of ChAdOx1 nCoV-19 vaccination and the immunological response after COVID-19 infection in ESRD patients with HD. The blood samples were obtained at baseline, 1-month, and 3-month follow-up after each shot or recovery. All participants were measured for anti-spike IgG by the ELISA method, using Euroimmun. This study found a significant increase in anti-spike IgG after 1 month of two-shot ChAdOx1 nCoV-19 vaccination, followed by a significant decrease after 3 months. On the other hand, the anti-spike IgG was maintained in the post-recovery group. There was no significant difference in the change of anti-spike IgG between the one-shot ChAdOx1 nCoV-19-vaccinated and post-recovery groups for both 1-month and 3-month follow-ups. The seroconversion rate for the vaccinated group was 60.32% at 1 month after one-shot vaccination and slightly dropped to 58.73% at the 3-month follow-up, then was 92.06% at 1 month after two-shot vaccination and reduced to 82.26% at the 3-month follow-up. For the recovered group, the seroconversion rate was 95.65% at 1 month post-recovery and 92.50% at 3-month follow-up. This study demonstrated the immunogenicity of two-dose ChAdOx1 nCoV-19 in ESRD patients with HD for humoral immunity. After COVID-19 infection, the humoral immune response was strong and could be maintained for at least three months.

## 1. Introduction

The first case of the Severe Acute Respiratory Syndrome Coronavirus 2 (SARS-CoV-2) was reported in Wuhan, China, in December 2019 as a novel causative pathogen of pneumonia with an unknown origin [1]. The novel coronavirus (COVID-19) has spread immensely worldwide, and COVID-19 was declared as a pandemic on 11 March 2020, by the World Health Organization (WHO) [2]. Various underlying comorbidities were associated with COVID-19 severity in hospitalized patients [3]. Chronic kidney disease patients had a high risk of COVID-19 severity and mortality [4,5], with a roughly three-fold risk of severe COVID-19 [6]. During the early phase of the pandemic, studies on end-stage renal disease (ESRD) patients with dialysis reported about one-third of the mortality of hospitalized patients with COVID-19 [7], with an estimated 8.7–12.9 excess mortality per 1000 ESRD patients [8]. A recent meta-analysis found a 4.3% prevalence of COVID-19, a 14.9% case fatality rate [9], and an 8.0% incidence in ESRD patients receiving hemodialysis (HD) [10].

ChAdOx1 nCoV-19 (AZD1222) is a chimpanzee adenovirus-vectored vaccine with a SARS-CoV-2 spike protein insertion. After a single dose, the ChAdOx1 nCoV-19 vaccine demonstrated acceptable safety and immunogenicity in terms of both antibody and T cell responses [11], with a stronger antibody response [12], but no T cell response after a booster dose [13]. ChAdOx1 nCoV-19 had similar immunogenicity across all age groups in healthy adults [14], and good clinical efficacy for the prevention of symptomatic COVID-19 infection for a two-dose regimen [15], with high efficacy at a three-month dose interval [16]. Recent studies reported reduced efficacy of ChAdOx1 nCoV-19 against the alpha variant [17] and no protection against the beta variant with mild-to-moderate symptoms [18].

Several studies have found that ESRD patients with HD had problems with vaccine efficacy for influenza [19,20], hepatitis B [21,22], and pneumococcal vaccine [23]. Similarly, studies on the mRNA COVID-19 vaccine in ESRD patients with HD reported a substantially reduced immune response after one dose [24] and two doses, particularly in older adults [25,26,27]. While promising results on immunogenicity and safety in immunocompromised hosts were reported in a trial of HIV patients receiving antiretroviral therapy [28], there was no study on ChAdOx1 nCoV-19 in ESRD patients with HD. We studied the immunogenicity of the ChAdOx1 nCoV-19 vaccination in ESRD patients with HD receiving a two-dose regimen with a 3-month dose interval.

## 2. Materials and Methods

### 2.1. Study Design and Population

In this prospective cohort study, we recruited 109 ESRD patients with HD, including 63 ESRD patients who were eligible for two doses of ChAdOx1 nCoV-19 vaccination with a 3-month interval between each dose, and 46 ESRD patients who recovered from COVID-19 infection in the national referral institute for infectious disease, Bamrasnaradura Infectious Diseases Institute in Nonthaburi, Thailand. Baseline demographic data were recorded using the case record form on the enrollment day. Blood samples were collected from all ESRD patients with HD for SARS-CoV-2 spike Immunoglobulin G (IgG) at baseline, 1 month, and 3 months after each dose of the ChAdOx1 nCoV-19 vaccination or infection with COVID-19.

The Institutional Review Board of the Bamrasnaradura Infectious Diseases Institute, Thailand (no. S010h/64; 2 June 2021) approved this study on ESRD patients with HD for the post-vaccination part, and the Ethics Committee of Research related to COVID-19 Disease or Public Health Emergency, Department of Disease Control, Ministry of Public Health, Thailand (no. 64061; 25 August 2021) approved this study on ESRD patients with HD for the post-COVID-19 infection part. Written informed consent was obtained from all participants before enrollment in the study.

### 2.2. Serum Collection for Antibody Testing

A blood sample of 10 mL from each participant was masked with a research identification number and sent to the Bamrasnaradura Infectious Disease Institute laboratory within 4 h and then centrifuged at 1500× *g* rpm for 5–10 min to separate serum. The serum was stored at −80 Celsius in the laboratory.

### 2.3. Serological Analysis for Humoral Immunogenecity

The SARS-CoV-2 QuantiVac ELISA (IgG) agent (Euroimmun, Lübeck, Germany) was used for SARS-CoV-2 spike IgG measurement. This method provided a quantitative result of the antibody against the spike protein as binding antibody units (BAU) per mL, which was recommended by the World Health Organization. Antibody levels of 32 BAU per mL or higher were considered positive for SARS-CoV-2 immunity.

### 2.4. Statistical Evaluation

Demographic data were presented using descriptive statistics, with continuous data presented as a mean with a 95% confidence interval and categorical data presented as counts and percentages. Differences in demographic data between two populations were analyzed using the χ^2^-test and Fisher’s exact test for categorical data and an unpaired *t*-test for continuous data, as appropriate. Differences between each participant were compared using a paired *t*-test. Subgroup analysis was conducted by population type (vaccinated versus recovered). The magnitude of the difference between the two subgroups was evaluated using an unpaired *t*-test. Graph visualizations were generated using GraphPad Prism version 9.0.0 for Windows (GraphPad Software, San Diego, CA, USA). Statistical analysis was conducted using STATA version 15 (College Station, TX, USA). A two-tailed *p*-value < 0.05 was considered statistically significant.

## 3. Results

### 3.1. Participant Characteristics

This prospective observational cohort study followed 109 ESRD patients with HD who either received a two-shot dose of ChAdOx1 nCoV-19 or were infected with SARS-CoV-2 for 3 months. The mean age of participants was 54.85 ± 15.23 years. Of 109 ESRD patients, 49 (44.95%) had diabetes mellitus type II, 90 (82.57%) had hypertension, and 4 (3.67%) had coronary artery disease. There were 63 ESRD patients who were eligible for the two-dose ChAdOx1 nCoV-19 vaccination and 46 ESRD patients who recovered from COVID-19 infection. All patients who recovered from COVID-19 infection had symptomatic COVID-19 infection and were hospitalized. The mean age of the vaccinated group was slightly higher at 57.63 ± 14.83 years, compared with the mean age of the recovered group at 51.13 ± 15.11 years. At baseline, the vaccinated group had a significantly higher age and a higher proportion of hypertension than the recovered group (Table 1).

### 3.2. Humoral Immunogenicity

A subgroup analysis by population type was conducted. Of 109 ESRD patients with HD, 63 were ESRD patients who received two doses of ChAdOx1 nCoV-19, and 46 were ESRD patients who recovered from COVID-19. In the vaccinated patients, the mean baseline anti-spike IgG level was 0 BAU/mL, which substantially increased to 66.13 BAU/mL (95% confidence interval (CI), 44.65–87.61) after receiving one dose of ChAdOx1 nCoV-19 at the 1 month follow-up (*p* < 0.001), then there was a slight decline to 54.97 BAU/mL (95% CI, 34.23–75.71) at 3 months post-one-shot vaccination (*p* = 0.279). Among the recovered patients, the mean baseline anti-spike IgG level was 4168.49 BAU/mL (95% CI, 2026.14–6310.84), which slightly decreased to 4117.94 BAU/mL (95% CI, 2015.44–6220.44) at 1 month post-infection (*p* = 0.960), and then gradually decreased to 3943.73 BAU/mL (95% CI, 1737.84–6149.61) at 3 months post-infection (*p* = 0.376). There was no significant difference in the change of anti-spike IgG after 1 month of follow-up between the partially vaccinated and recovered groups (*p* = 0.907). Similarly, after 3 months of follow-up, there was inadequate evidence for a significant difference in anti-spike IgG levels (*p* = 0.368) (Figure 1). The seroconversion rate for ESRD patients with HD who received one dose of ChAdOx1 nCoV-19 was 60.32% at 1-month post-vaccination and slightly dropped to 58.73% at the 3-month follow-up. For ESRD patients with HD who recovered from COVID-19 infection, the seroconversion rate was 95.65% at 1-month post-recovery and 92.50% at the 3-month follow-up.

In the vaccinated ESRD with HD patients, the second shot of ChAdOx1 nCoV-19 was administered at a 3-month follow-up after taking blood samples. The mean anti-spike IgG level before the second shot was 54.97 BAU/mL, which substantially increased to 435.96 BAU/mL (95% CI, 287.18–584.74) after receiving two doses of ChAdOx1 nCoV-19 at the 1-month follow-up (*p* < 0.001), then there was a decline to 260.74 BAU/mL (95% CI, 173.90–347.59) at 3 months after the two-shot vaccination (*p* < 0.001) (Figure 2). The seroconversion rate for ESRD patients with HD who received two doses of ChAdOx1 nCoV-19 was 92.06% at 1-month post-vaccination and reduced to 82.26% at the 3-month follow-up.

## 4. Discussion

This prospective cohort study evaluated the immunogenicity of ChAdOx1 nCoV-19 as a second-dose vaccination and humoral immunity following COVID-19 infection in ESRD patients with HD. Anti-spike IgG levels were considerably increased one month following two ChAdOx1 nCoV-19 vaccinations but then dropped three months later. Meanwhile, in the post-recovery group, the anti-spike IgG level remained relatively stable. For both 1-month and 3-month follow-ups, there was no difference in anti-spike IgG change between the vaccinated and recovered groups.

Our results show that anti-spike IgG levels were considerably greater 1 month after each dosage of ChAdOx1 nCoV-19 in ESRD patients receiving hemodialysis, which was consistent with humoral responses following a single dose of ChAdOx1 nCoV-19 [29,30], and two doses of ChAdOx1 nCoV-19 [31,32]. ChAdOx1 nCoV-19 appeared to have variable immunogenicity in patients with varying degrees of immunocompromised status. Notably, ChAdOx1 nCoV-19 has shown promising immunogenicity in ESRD patients with HD compared with those undergoing kidney transplantation [33]. Similarly, ChAdOx1 nCoV-19 did not work well in heart transplant recipients [34]. On the other hand, patients with HIV had favorable immune responses after two doses of ChAdOx1 nCoV-19 [35]. While the humoral immunogenicity of two doses of ChAdOx1 nCoV-19 one month after vaccination was relatively higher in ESRD with HD patients with a mean anti-Spike IgG level of 435.96 BAU/mL compared to the healthy population with 173.62 BAU/mL, the seroconversion rate was lower in ESRD with HD patients, with 92.06% compared to the healthy population with 99.00–100.00% [36,37].

In this study, we exhibited significant immunogenicity of the second dose of ChAdOx1 nCoV-19 after the 1-month and 3-month follow-ups. However, the seroconversion rate after one dosage of ChAdOx1 nCoV-19 in ESRD with HD patients was not ideal, and the gap between the first and second dose was relatively considerable. Thus, during ongoing pandemics, ChAdOx1 nCoV-19 may not be an ideal alternative for ESRD with HD patients in particular cases where SARS-CoV-2 protection is urgently required. Meanwhile, the results for mRNA COVID-19 vaccines, including BNT162b2 and mRNA-1273, were inconclusive. Some studies claimed that BNT162b2 had favorable immunogenicity in ESRD patients on hemodialysis [38,39,40,41], while others found it to have low yet adequate immunogenicity [26,27,42,43,44,45,46,47]. A multicenter cohort study in ESRD with hemodialysis patients found that two doses of mRNA-1273 were more immunogenic than BNT162b2 [48,49]. In addition, the immunogenicity of CoronaVac was very low after one month of two-dose vaccination, with only 44 percent seroconversion, but it reached a peak at three months of two-dose vaccination, with 86.5 percent seroconversion [50].

Our results showed that the IgG level after COVID-19 infection in ESRD patients with HD was comparable with earlier studies on relatively high IgG levels [51,52], but not with previous studies on IgG dynamics, which suggested a considerable reduction in the IgG level during 4 months of follow-up [53]. Similar to the findings in ESRD patients with HD after COVID-19 infection, the results in the general population on the dynamics of IgG levels post-COVID-19 infection were inconclusive. Some studies in the general population reported a substantial decline in IgG [54] and neutralizing antibody over 3 months post-infection [55], yet other studies showed a stable antibody level after 6 [56,57,58] to 12 months post-infection [59]. Interestingly, the dynamics of IgG levels following infection in our study were similar to those in ESRD patients with kidney transplantation, who had significant IgG persistence over 3 months after COVID-19 infection [60]. It is also notable that, in HIV patients, similar persistence of post-infection IgG levels was reported over a three-month period [61]. The IgG seroconversion rate was slightly lower in ESRD with HD patients, with 96% at 1-month post-recovery and 93% at 3-month follow-up compared with 96% at both 6- and 12-month follow-up in the healthy population [59].

The vaccinated group had a lower absolute mean anti-spike IgG level than the recovered group, as well as a lower seroconversion rate. The higher anti-spike IgG level in the recovered group could be attributed to the recovered group’s eligibility criteria, which comprised only symptomatic and hospitalized COVID-19 patients with a higher humoral immune response than asymptomatic COVID-19 patients [62]. Although there was a significant difference in age between the vaccinated and recovered groups, there was evidence that, in the general population, age was not associated with IgG response to ChAdOx1 nCoV-19 [37]. Concerning comorbidity, there was evidence that the population with comorbidity might acquire a greater serum IgG level. With the majority of the vaccinated group having comorbidity, the serum IgG level might be somewhat overstated in ESRD with HD patients without comorbidity [36]. Surprisingly, the results contradicted a general population study that evaluated serum IgG levels following COVID-19 recovery and a two-dose mRNA vaccination, which found a greater serum IgG level in the vaccinated group compared to the recovered group [63]. The difference in results could be attributable to the different vaccine types or the population’s immunological condition. 

This study has several strengths. To the best of our knowledge, this was the first cohort study in the ESRD with HD population that compared patients who were vaccinated with two doses of ChAdOx1 nCoV-19 and recovered from COVID-19 infection. In addition, there were multiple follow-ups, allowing for the observation of IgG level variations over time, and the immunogenicity information in this study was derived from individual data. Furthermore, the sample recruited in this study was a high-risk population with immunocompromised status whose immune response may differ from that of the general population.

The limitations in our study include the incomplete follow-up of ESRD patients who were recovered from COVID-19 at 3 months post-infection, which caused the results to appear contradictory when comparing the anti-spike IgG immunogenicity of 1 and 3 months post-recovery. Drop-out participants tended to exhibit relatively higher anti-spike IgG levels at 1-month follow-up compared with those who remained in the study. Another limitation was that we focused on the humoral immune response, which may not explain the entire immune response in ESRD patients with HD. Furthermore, due to a lack of scientific evidence and resources, this investigation concentrated on IgG levels. As a result, no additional immunoglobulin tests were planned or carried out. Given the rise in the delta variant during the study period, the inclusion of primarily symptomatic and hospitalized COVID-19 patients in the recovered group may overestimate blood IgG levels. Furthermore, since this study was conducted in a low-to-middle-income country with limited vaccine resources, the vaccination policy may not reflect the current vaccination policy in some countries, and even now, the vaccination interval for ChAdOx1 nCoV-19 is unchanged [64], while the World Health Organization still recommends an interval of 8 to 12 weeks [65]. The final constraint was the relatively short follow-up time, which prevented us from observing the time point at which anti-spike IgG levels in the post-infection group began to fall dramatically.

## 5. Conclusions

Taken together, this study established the immunogenicity of two-shot ChAdOx1 nCoV-19 in ESRD patients with HD for humoral immunity. After COVID-19 infection, the humoral immune response was strong and could be maintained for at least three months.

## Figures and Tables

**Figure 1 vaccines-10-00959-f001:**
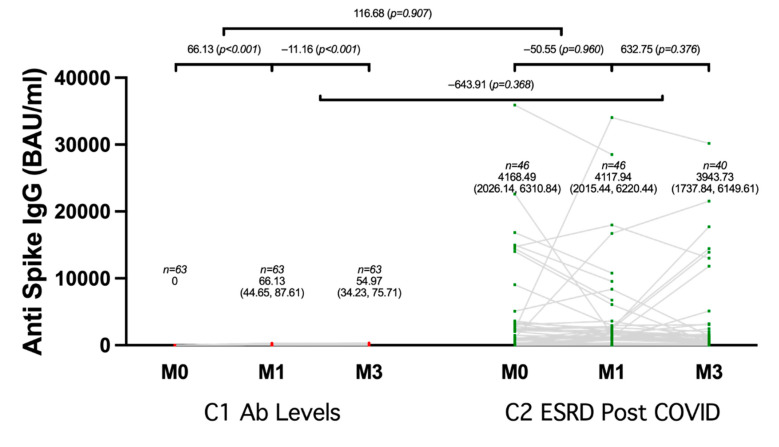
Changes in anti-spike IgG in 109 ESRD patients, including 63 who received the first dose of a two-dose regimen of ChAdOx1 nCoV-19, and 46 who recovered from COVID-19, were assessed at baseline (M0), 1 month, and 3 months. BAU stands for binding antibody units, and IgG stands for immunoglobulin G.

**Figure 2 vaccines-10-00959-f002:**
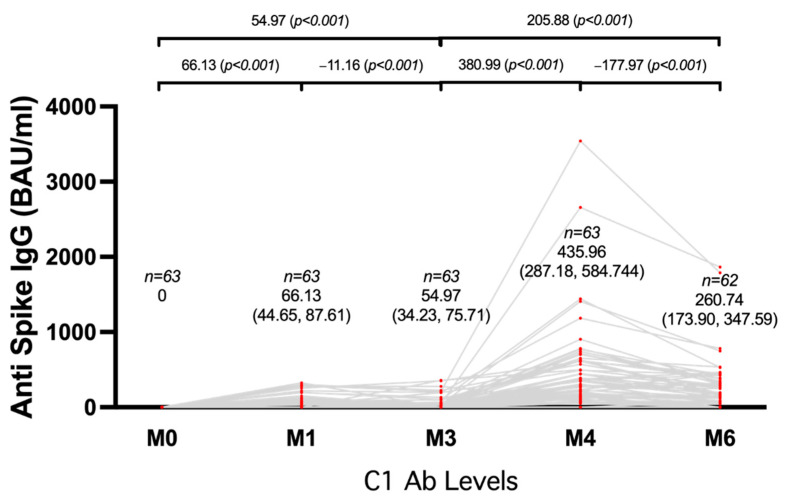
Changes in anti-spike IgG were evaluated at baseline (M0), 1 month, and 3 months after each shot in 63 ESRD with HD patients who received two shots of ChAdOx1 nCoV-19, and is presented as BAU/mL. BAU stands for binding antibody units, and IgG stands for immunoglobulin G.

**Table 1 vaccines-10-00959-t001:** Demographic data of participants (*n* = 109).

Characteristics	Total	Vaccinated	Recovered	*p*-Value
Participants	109	63	46	
Age, years (mean ± SD)	54.93 ± 15.28	57.63 ± 14.83	51.22 ± 15.26	0.03
BMI, kg/m^2^ (mean ± SD)	23.86 ± 4.80	24.61 ± 5.19	22.83 ± 4.03	0.06
Comorbidities	94 (86.24%)	57 (90.48%)	37 (80.43%)	0.16
Diabetes mellitus	49 (44.95%)	33 (52.38%)	16 (34.78%)	0.08
Hypertension	89 (81.65%)	56 (88.89%)	33 (71.74%)	0.03
Coronary artery disease	4 (3.67%)	3 (4.76%)	1 (2.17%)	0.64

Data are presented as counts and percentages if not otherwise specified. BMI: body mass index; SD: standard deviation.

## Data Availability

The supporting data for the findings of this study are available from the corresponding author on reasonable request.

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
