# Peer review of "ChAdOx1 nCoV-19 Immunogenicity and Immunological Response Following COVID-19 Infection in Patients Receiving Maintenance Hemodialysis"

_vaccines, 2022, doi:10.3390/vaccines10060959_

Round 1

Reviewer 1 Report

In this study the authors recruited 109 ESRD patients with HD, including 81 63 ESRD patients who were eligible for the two-dose of ChAdOx1 nCoV-19 with a 3-month 82 interval between each dose, and 46 ESRD patients who recovered from COVID infection, to evaluate the immunogenicity of ChAdOx1 (2 doses) in comparison with natural infection.
The study is well designed and manuscript is readable and well written. The conclusions are in agreement with the results shown.

Author Response

Reviewer 1: In this study the authors recruited 109 ESRD patients with HD, including 63 ESRD patients who were eligible for the two-dose of ChAdOx1 nCoV-19 with a 3-month interval between each dose, and 46 ESRD patients who recovered from COVID infection, to evaluate the immunogenicity of ChAdOx1 (2 doses) in comparison with natural infection. The study is well designed and manuscript is readable and well written. The conclusions are in agreement with the results shown.

Response: Thank you for your supportive comments. We believe this study provides useful evidence for COVID -19 vaccination and pandemic control in the unique patient population.

Reviewer 2 Report

I've read with interest the present paper from the study group of Prasithsirikul et al. They analyzed the immunogenicity of two doses of ChAdOx1 nCoV-19 and the immune 28 response post-COVID-19 infection in ESRD with HD patients. These results support the use of vaccination in HDP patient to reduce the high mortality rate in this subgroup. They observed a significant seroconversion in HD patients following vaccine administration or COVID-19 infection. Notably, vaccination or covid-19 infection induce an increase in anti-spike IgG after 1 month of two-shot ChAdOx1 nCoV-19 vaccination, followed by a significant decrease after 3 months. These results support the use of vaccination in HDP patient to reduce the high mortality rate in this subgroup. However, the manuscript has some important weaknesses which must be addressed before any consideration.

  • The format of the manuscript: data presented in this research are not enough for a full-length article, authors may consider other formats such as letter to editor etc.
  • The lack of a good control population reduces the significance of the research. Because of this limitation, we encourage the authors to provides more details on the HDP vaccinated population including symptomatic and non-symptomatic infection post vaccination; the occurrence of side effects etc.
  • In addition, we encourage authors to compare all the data with other published data on healthy people.
  • Sample isolation: description of serum isolation and storage is needed
  • Comorbidities are highly relevant for covid-19 immunization and recovery; the reviewer is worried by the significant difference in comorbidities between vaccinated group and the recovered group. Could this explain why the seroconversion is high in recovered patients compared to vaccinated ones.
  • Data presented revealed that recovered group displayed at least ten time the level of IgG detected in the vaccinated group. Please discuss this difference with regard to other published studied on immunization and post recovery.
  • Any data about other immunoglobulins. IgG indeed is widely used to analyze seroconversion in patients. However recent data provide evidence that, it is also worthy to analyze IgM. In case that IgM
  • The goal and the conclusion of the study are not clear, this should clearly appear in the abstract.
  • Figure 1, legend: “63 who received…a single dose….”. Early, it is written that the 63 patients received “2 doses”, please could you clarify the number of doses?
  • Figure F2, it seems like figure 2 is a part of figure 1. If, it is the case the authors should present only one figure or 2 figures with different data.

Author Response

Reviewer 2: I've read with interest the present paper from the study group of Prasithsirikul et al. They analyzed the immunogenicity of two doses of ChAdOx1 nCoV-19 and the immune response post-COVID-19 infection in ESRD with HD patients. These results support the use of vaccination in HDP patient to reduce the high mortality rate in this subgroup. They observed a significant seroconversion in HD patients following vaccine administration or COVID-19 infection. Notably, vaccination or covid-19 infection induce an increase in anti-spike IgG after 1 month of two-shot ChAdOx1 nCoV-19 vaccination, followed by a significant decrease after 3 months. These results support the use of vaccination in HDP patient to reduce the high mortality rate in this subgroup. However, the manuscript has some important weaknesses which must be addressed before any consideration.

Response: Thank you for your time and consideration. The manuscript was revised to address the concerning issues.

Reviewer 2: The format of the manuscript: data presented in this research are not enough for a full-length article, authors may consider other formats such as letter to editor etc.

Response: Thank you for providing an alternative option. We feel that the "letter to the editor" format may not convey all the important information and messages of the study in a limited space. In addition, more data and discussion have been added to warrant a full-length article.

Reviewer 2: The lack of a good control population reduces the significance of the research. Because of this limitation, we encourage the authors to provides more details on the HDP vaccinated population including symptomatic and non-symptomatic infection post vaccination; the occurrence of side effects etc.

Response: Thank you for bringing this to our attention. The study's goal for the vaccinated HDP cohort was to assess the immunogenicity of ChAdOx1 nCoV-19 immunization in ESRD patients with HD. As a result, no clinical follow-up was scheduled following vaccination. While a robust control population was crucial in an analytic study, we feel the study was important in epidemiologic studies in a unique population for whom no scientific data exists, and it provided critical evidence for evidence-based practice and policy implementation.

Reviewer 2: In addition, we encourage authors to compare all the data with other published data on healthy people.

Response: Thank you for your suggestions. The data were compared with other published data on healthy populations accordingly in the Discussion section.

Reviewer 2: Sample isolation: description of serum isolation and storage is needed

Response: The description of serum isolation and storage was provided accordingly in the Methods section.

Reviewer 2: Comorbidities are highly relevant for covid-19 immunization and recovery; the reviewer is worried by the significant difference in comorbidities between vaccinated group and the recovered group. Could this explain why the seroconversion is high in recovered patients compared to vaccinated ones.

Response: Statistical tests were performed to evaluate the difference in comorbidities between the vaccinated group and the recovered group. Additional discussion of seroconversion was provided accordingly in the Discussion section.

Reviewer 2: Data presented revealed that recovered group displayed at least ten time the level of IgG detected in the vaccinated group. Please discuss this difference with regard to other published studied on immunization and post recovery.

Response: Additional discussion on the level of IgG was provided in the Discussion section accordingly.

Reviewer 2: Any data about other immunoglobulins. IgG indeed is widely used to analyze seroconversion in patients. However recent data provide evidence that, it is also worthy to analyze IgM. In case that IgM

Response: Thank you for your insightful recommendation. This study aimed to examine the immunogenicity of ChAdOx1 nCoV-19 immunization and the immune response after COVID-19 infection in ESRD patients with HD. No additional immunoglobulin testing was planned due to scientific evidence, clinical experience during the research period, and limited resources. In the Discussion section, this limitation was addressed. We could not understand your last phrase (“In case that IgM”) which seemed to be an incomplete sentence.

Reviewer 2: The goal and the conclusion of the study are not clear, this should clearly appear in the abstract.

Response: The abstract was revised to provide a more clear and concise goal and conclusion of the study accordingly.

Reviewer 2: Figure 1, legend: “63 who received…a single dose….”. Early, it is written that the 63 patients received “2 doses”, please could you clarify the number of doses?

Response: We apologize for the confusion. All 63 patients got two doses of ChAdOx1 nCoV-19, while Figure 1 compares the immunological response following the first dosage of ChAdOx1 nCoV-19 and those with COVID -19. The information has been revised accordingly.

Reviewer 2: Figure F2, it seems like figure 2 is a part of figure 1. If, it is the case the authors should present only one figure or 2 figures with different data.

Response: While Figure 2 provides some of the information shown in Figure 1 for M0-M3, we believe that Figure 2 provides additional important information on IgG levels after vaccination for the second dose of ChAdOx1 nCoV-19 with the correct scale to more clearly show the dynamics of IgG levels.

Reviewer 3 Report

This paper describe the effecs of two doses vaccination in an Hemodialysis population. Authors also made a comparison Into the studied population between vaccinated and recovered from infection.

The results showed by authors are not novel and are related to an old vaccination policy (i.e two doses with 3 month interval)

I think that those infomation are not of clinical help in the current clinical practice 

Author Response

Reviewer 3: This paper describe the effecs of two doses vaccination in an Hemodialysis population. Authors also made a comparison Into the studied population between vaccinated and recovered from infection. The results showed by authors are not novel and are related to an old vaccination policy (i.e two doses with 3 month interval). I think that those infomation are not of clinical help in the current clinical practice.

Response: Thank you for highlighting the issues. Although the results and methodology for serologic testing are not novel, we conducted the study in a unique population with severe COVID-19 and no published evidence. This study, we believe, will provide useful information for practice and policy implementation. Concerning old vaccination practices, each country's immunization policies have been unique and dynamic. Immunization policies in low- and middle-income nations were dependent on resource availability. Please see the Discussion section for more details.

We feel that this information is critical for individuals with end-stage renal disease with HD due to the lack of data for this patient population. In addition, current therapeutic practices vary from country to country due to the availability of resources and the execution of policies. Information that seemed unhelpful in one country could save multiple lives in another.

Round 2

Reviewer 2 Report

Even though the reviewer still believe that an letter for editor is the more suitable format for this study, he agrees with the author that the description of the Covid-19 immunization in ESRD with HD patients is quite original.

Author Response

Reviewer: Even though the reviewer still believe that an letter for editor is the more suitable format for this study, he agrees with the author that the description of the Covid-19 immunization in ESRD with HD patients is quite original.

Response: Thank you so much for the comment.

Reviewer 3 Report

The authors modified the paper , but unfortunately i don't think this modify my opinion since in my view the paper is not adding any information to the already available literature, and the population is not big enough to confirm previous observations . The authors mentioned no literature in the field of immunity after covid infection in  ESRD patients, I don't agree,  for example a wide series of cases (356 patients) were described in  "Longevity of SARS-CoV-2 immune responses in haemodialysis patients and protection against reinfection"  KI 2021 Jun;99(6):1470-1477.  Therefore , even if the paper improved in several aspect after major revisions I still don’t feel the paper is novel.

Author Response

Reviewer: The authors modified the paper , but unfortunately i don't think this modify my opinion since in my view the paper is not adding any information to the already available literature, and the population is not big enough to confirm previous observations . The authors mentioned no literature in the field of immunity after covid infection in  ESRD patients, I don't agree,  for example a wide series of cases (356 patients) were described in  "Longevity of SARS-CoV-2 immune responses in haemodialysis patients and protection against reinfection"  KI 2021 Jun;99(6):1470-1477.  Therefore , even if the paper improved in several aspect after major revisions I still don’t feel the paper is novel.

Response: We appreciate the reviewer's remark highlighting a potential weakness in our study. We have likewise reviewed with great interest the series published in the KI journal referenced by the reviewer and have compared the results to our own. While there are parallels between the two papers, we feel ours contributes valuable non-Caucasian data from a developing country. Although Thailand was one of the first countries to experience COVID-19, our study was launched later than desired due to a lack of funding and ethical approval, as well as the need to collect sufficient cumulative cases. This is why we feel novelty may not be a criterion for publishing in Vaccines. Given the restricted resources available, we feel the concept and execution of our study match international publishing standards.